# Size and Shape Selective Classification of Nanoparticles

**Cornelia Damm** [1], **Danny Long** [1,2], **Johannes Walter** [1,2] and **Wolfgang Peukert** [1,2,*]

[1] Institute of Particle Technology (LFG), Friedrich-Alexander-Universität Erlangen-Nürnberg (FAU), Cauerstr. 4, 91058 Erlangen, Germany; cornelia.damm@fau.de (C.D.); danny.long@fau.de (D.L.); johannes.walter@fau.de (J.W.)

[2] Center for Functional Particle Systems, Friedrich-Alexander-Universität Erlangen-Nürnberg (FAU), Haberstr. 9a, 91058 Erlangen, Germany

[*] Correspondence: wolfgang.peukert@fau.de; Tel.: +49-9131-85-29400

**Abstract:** As nanoparticle syntheses on a large scale usually yield products with broad size and shape distributions, the properties of nanoparticle-based products need to be tuned after synthesis by narrowing the size and shape distributions or via the removal of undesired fractions. The development of property-selective classification processes requires a universal framework for the quantitative evaluation of multi-dimensional particle fractionation processes. This framework must be applicable to any property and any particle classification process. We extended the well-known one-dimensional methodology commonly used for describing particle size distributions and fractionation processes to the multi-dimensional case to account for the higher complexity of the property distribution and separation functions. In particular, multi-dimensional lognormal distributions are introduced and applied to diameter and length distributions of gold nanorods. The fractionation of nanorods via centrifugation and by orthogonal centrifugal and electric forces is modeled. Moreover, we demonstrate that analytical ultracentrifugation with a multi-wavelength detector (MWL-AUC) is a fast and very accurate method for the measurement of two-dimensional particle size distributions in suspension. The MWL-AUC method is widely applicable to any class of nanoparticles with size-, shape- or composition-dependent optical properties. In addition, we obtained distributions of the lateral diameter and the number of layers of molybdenum disulfide nanosheets via stepwise centrifugation and spectroscopic evaluation of the size fractions.

**Keywords:** multi-dimensional particle size distribution; multi-dimensional separation; analytical ultracentrifugation with multi-wavelength detector





## 1. Introduction

The properties of particles are, in general, governed by their distributions in terms of size, shape, structure, composition, surface and functional properties. A comprehensive description and characterization of this multi-dimensional property space is currently not possible. Nevertheless, a clear trend towards increasing complexity of functional particle systems is evident. For instance, a large number of protocols for size- and shape-selective nanoparticle synthesis on a small scale exists. However, the scale-up of these methods inevitably leads to the broadening of the particle size and shape distribution because the mixing and reaction conditions in larger reactors are less uniform than in small vessels. Thus, for large-scale nanoparticle syntheses, a subsequent separation step is necessary for tuning the product's properties according to the requirements of the specific product application by narrowing the size and shape distribution or via the removal of undesired coarse or fine fractions. Traditional methods for size classification of particles on an industrial scale, like cyclones and deflector wheel classifiers, rely on mass forces acting on the particles. These methods are limited to particles with dimensions in the micrometer range, because the mass forces acting on nanoparticles are too weak, so that nanoparticles simply follow the fluid flow without any separation regarding size and shape.

Although the separation of nanoparticles is still challenging, highly promising results for classification regarding size were obtained by using methods like nanoparticle chromatography [1–8], size-selective precipitation of semiconductor quantum dots [9,10], electrophoresis [11–14] and field flow fractionation [15–22]. In particular, chromatographic methods for the classification of nanoparticles are promising because of their high efficiency and good scalability. Material-independent size classification of spherical nanoparticles with diameters between 5 and 80 nm using size-exclusion chromatography with a separation sharpness of up to 0.87 (analytical separation) was demonstrated [4]. ZnS quantum dots were classified regarding band gap energy via size-exclusion chromatography, with a separation efficiency of about 0.9 (sharp analytical separation) and a yield of up to 72% for the fine fraction [2]. Anion-exchange chromatography was used to classify gold clusters < 2 nm, and clusters consisting of 10, 15, 25 and 29 gold atoms, respectively, were isolated as single components [3]. Centrifuges providing sufficiently high centrifugal forces (g-value) to separate nanoparticles < 100 nm at a scale of some hundreds of milliliters or even at liter scale become now available [23–25]. The material-specific classification of nanoparticles 20–100 nm in diameter can be achieved via magnetic field-assisted methods like porous ferromagnetic membranes [26] or magnetic field chromatography [27] but require either magnetic nanoparticles or a selective binding of magnetic particles to the desired nanoparticle fraction. Shape-selective classification of nanoparticles was realized in lab-scale via gel electrophoresis (separation of gold nanorods with a length < 70 nm from spherical gold nanoparticles) [28] and density-gradient ultracentrifugation. Examples of the latter are the classification of 2D-nanosheets (50 to few hundreds nm in diameter, thickness few nm) with respect to the layer thickness, where graphene monolayers with a purity of 80% were obtained [29–31], the separation of gold nanorods with lengths of about 100 nm from about 20 nm spherical and cubic gold nanoparticles [12] and the classification of single-wall carbon nanotubes regarding length and chirality [32,33].

Promising methods for multi-dimensional particle fractionation regarding size, shape and other properties were developed by several projects within a national priority program (PP 2045—highly specific and multi-dimensional fractionation of particle systems with technical relevance) [34,35] in Germany. At the beginning of PP 2045, it was not clear how to describe multi-dimensional particle size distributions (PSDs) and fractionation processes quantitatively. Moreover, highly precise ensemble methods for measuring multi-dimensional particle size distributions in suspension, i.e., with good statistics, are rare. The projects of U. Peuker [36,37] and our project within PP 2045 contributed to a deeper understanding of such complex multi-dimensional separation processes by developing a uniform mathematical framework for the quantitative description of multi-dimensional particle property distributions and multi-dimensional fractionations. For that purpose, we extended the well-known approach for describing one-dimensional PSDs and fractionations to the multi-dimensional normal distributed case [38–40]. In PP 2045, we further developed analytical ultracentrifugation with UV/Vis/NIR multi-wavelength detector (MWL-AUC) as a fast and highly accurate ensemble method for measuring multi-dimensional particle property distributions in suspension [39–42]. Moreover, in the first funding period of PP 2045, we demonstrated the feasibility of nanoparticle chromatography for plasmonic gold nanoparticles [1] and semiconducting quantum dots [2].

This paper is the final report of our project in PP 2045. Beyond our already published results [38–41], in this paper, we extend our uniform framework to higher-dimensional lognormal distributions and use it, for illustration, to describe the two-dimensional distribution of gold nanorods. Additionally, we calculate the outcome of one- and two-dimensional fractionation processes applied to gold nanorods with lognormal distributions of length and diameter. Moreover, we demonstrate the measurement of two-dimensional PSDs by MWL-AUC exemplary for gold nanorods [39,40,42] and present a simple method for measuring size distributions of molybdenum disulfide nanosheets based on preparative centrifugation.

## 2. Materials and Methods

### 2.1. Materials

An aqueous suspension of gold nanorods (product number: A12-25-750-CIT-DIH-1-25; LOT #: N2294) stabilized by trisodium citrate was purchased from Nanopartz™. Molybdenum disulfide < 2 μm (purity > 99%) from Sigma-Aldrich (St. Louis, MO, USA) was used as feed material for the preparation of molybdenum disulfide nanosheets. Sodium cholate > 99% (Calbiochem Merck, Darmstadt, Germany) was used as a stabilizing agent for the molybdenum disulfide nanosheets. All materials were used as received. Ultrapure water was used for the preparation of all suspensions.

### 2.2. Preparation and Classification of Molybdenum Disulfide Nanosheets

Nanosheets were prepared via stirred media delamination of the layered material molybdenum disulfide using a lab-scale batch stirred media mill "PE075" (Netzsch Feinmahltechnik GmbH, Selb, Germany). This method relies on shear forces, which are induced due to the interaction of the layered material with delamination beads, to overcome van der Waals interactions between the layers [43–45]. A total of 200 mL of ultrapure water containing 1 wt.% of molybdenum disulfide and 0.5 wt.% of sodium cholate was processed at 15 °C for 5 h with a stirrer rotation speed of 1000 rpm. Yttria-stabilized zirconia beads (100 μm in diameter, total mass 1.5 kg) were used as delamination media. After separating the molybdenum disulfide suspension from the delamination beads using a sieve with a mesh width of 64 μm, the sheets were classified regarding size using a preparative high-speed centrifuge "3-30KS" (Sigma Laborzentrifugen GmbH, Osterode, Germany). In the first classification step, the suspension was centrifuged to a sphere-equivalent cut size of 300 nm (4000 rpm/relative centrifugal force (rcf) of 1664× $g$ for 5 min) to separate the nanosheets from the not-yet-delaminated particles. The supernatant was collected and labeled as "fraction-300 nm". To obtain different size fractions of the nanosheets, the "fraction-300 nm" sample was centrifuged to sphere-equivalent cut sizes of 250 nm, 200 nm, 150 nm, 100 nm, 75 nm and 50 nm, respectively. The mentioned cut sizes correspond to the following centrifugation conditions: 4800 rpm/2396× $g$, 6000 rpm/3743× $g$ and 8000 rpm/6654× $g$ for each 5 min and 12,000 rpm/14,972× $g$ for 5 min, 9 min and 20 min, respectively. The supernatant of each centrifugation step was collected and labeled as "fraction-250 nm" up to "fraction-50 nm".

### 2.3. Analysis of the Two-Dimensional Nanoparticle Distributions

UV/Vis spectra were recorded using a spectrophotometer "Cary-100" (Varian, Palo Alto, CA, USA) and disposable cuvettes with an optical pathway of 1 cm.

Scanning transmission electron microscopy (STEM) experiments were performed using a probe-corrected Spectra 200 C-FEG TEM operated in STEM mode (Thermo Fisher Scientific Inc, Waltham, MA, USA) using a high-angle annular dark-field detector (collection-angle ranging from 56–200 mrad) with an acceleration voltage of 200 kV. Particles were drop-cast on a carbon film coated on a carrier mesh copper TEM grid from Plano GmbH, Wetzlar, Germany, with a specification of 200 mesh. The software ImageJ (version 1.43u) [46] was used for the analysis of the STEM images. Manual measurements using the straight-line tool on the obtained images were conducted to determine the lengths and diameters of the particles.

A preparative ultracentrifuge "Optima L-90K" (Beckman Coulter Life Sciences, Indianapolis, IN, USA) equipped with UV/Vis/NIR multi-wavelength optics (Nanolytics Instruments GmbH, Potsdam, Germany) and a "Flame-S-VIS-NIR" spectrometer (Ocean Optics Germany GmbH, Ostfildern, Germany) was used for the size analysis of two-dimensional nanoparticles in suspension. The samples were placed in titanium center-pieces with an optical path length of 12 mm, as well as in sapphire windows (Nanolytics Instruments GmbH, Potsdam, Germany). The sedimentation of the particles was investigated at 20 °C and a constant rotor speed of 3000 rpm using an An-60 Ti analytical rotor (Beckman Coulter Life Sciences, Indianapolis, IN, USA). The extinction of the suspensions

was measured at the radial position of 6.9 cm. Details of the setup, as well as the data acquisition, are described in our previous work [47,48]. Extinction-weighted sedimentation coefficient distributions were derived using a direct boundary model for dynamic rotor speed experiments [49].

### 3. Results

*3.1. Introduction of Mathematical Terminology*

In the following sections, we will describe various ways to mathematically describe the PSD one observes in measurement data or models. To prepare for this discussion, we first introduce the important terminology that we use in our work and explain the reasoning behind these approaches. It is helpful to distinguish between density and cumulative representations. PSDs are described with probability density functions (PDFs)—typically denoted as $q\left(\vec{x}\right)$, where $\vec{x}$ is the relevant property vector in terms of size, shape and other properties—which are functions that satisfy the following conditions:

$$q\left(\vec{x}\right) \geq 0 \text{ for all } \vec{x}, \tag{1}$$

$$\int q\left(\vec{x}\right)\mathrm{d}\vec{x} = 1. \tag{2}$$

This means that PSDs are normalized such that the area under the density distribution is equal to one. The integral in Equation (2) is performed over all "relevant properties", which, in the most general way, means values between zero and infinity for each relevant property description (e.g., diameter and length), but it can also have alternative upper and lower bounds based on the range of values observed in the data.

The second important presentation of PSDs is the cumulative distribution function (CDF), which describes the proportion of particles below a certain size relative to the total amount of particles. In the one-dimensional setting, this is defined as

$$Q(x) = \frac{amount\ of\ particles\ \leq\ x}{total\ amount} = \int_0^x q(z)\mathrm{d}z \tag{3}$$

In the two-dimensional setting, the CDF is defined as

$$Q(x,y) = \int_0^x \int_0^y q(z,w)\mathrm{d}z\mathrm{d}w. \tag{4}$$

For higher-dimension particles, the CDF can be defined in a similar manner.

With the CDF, the relative proportion of particles between sizes $\vec{x}$ and $\vec{y}$, where $\vec{x}_i < \vec{y}_i$ in each size component, can be computed in a straightforward manner with the CDF $Q\left(\vec{x}\right)$ given by Equation (5)

$$\text{Proportion of particles between sizes } \vec{x} \text{ and } \vec{y}: \Delta Q\left(\vec{x},\vec{y}\right) = Q\left(\vec{y}\right) - Q\left(\vec{x}\right). \tag{5}$$

In this work, we will focus on describing particles with a lognormal distribution, since this distribution has been found to describe a variety of different particulate processes including the size and velocity distributions of particles in two-phase flows [50], the size distributions of aerosols [51] and the size distributions of ultrasmall gold nanoparticles [52]. The lognormal distribution in *N* dimensions is defined as

$$q\left(\vec{x}\right) = (2\pi)^{-\frac{N}{2}}|\Sigma|^{-\frac{1}{2}}\left(\prod_{i=1}^N x_i^{-1}\right)\exp\left[-\frac{1}{2}\left(\ln\left(\vec{x}\right) - \mu\right)^{\mathrm{T}}\Sigma^{-1}\left(\ln\left(\vec{x}\right) - \mu\right)\right], \tag{6}$$

where $\mu$ is the mean value of $\ln\left(\vec{x}\right)$, $\Sigma$ is the covariance matrix of $\ln\left(\vec{x}\right)$ and $|\Sigma|$ indicates the determinant of the matrix $\Sigma$. For details regarding this notation, we refer to the Supplementary Materials.

The reason that the mean and covariance matrix are defined in terms of the logarithm of the particle sizes is because the lognormal distribution is defined such that the logarithm of $\vec{x}$ follows a normal distribution. The covariance matrix is very similar to the variance of a one-dimensional distribution, but it also accounts for the correlation between the variables. Mathematically, this is described with

$$\sigma_{xy} = E\left[(x - \mu_x)(y - \mu_y)\right], \tag{7}$$

where $\mu_x$ and $\mu_y$ indicate the mean values of variables $x$ and $y$, respectively, and $E[\cdot]$ indicates the mean value of the distribution weighted by the variable or function within the brackets. When $x = y$, then Equation (7) is equivalent to the familiar variance $\sigma_x^2$. It also follows from Equation (7) that $\sigma_{xy} = \sigma_{yx}$. The covariance terms $\sigma_{xy}$ indicate the relationship between variables $x$ and $y$. If $\sigma_{xy} > 0$, then an increase in $x$ is correlated with an increase in $y$; if $\sigma_{xy} < 0$, then an increase in $x$ is correlated with a decrease in $y$; if $\sigma_{xy} = 0$, then the values of $x$ and $y$ are uncorrelated. In Figure 1, we show examples of two-dimensional lognormal distributions with positive, negative and no correlation between the variables to demonstrate how the covariance matrix affects a PSD.

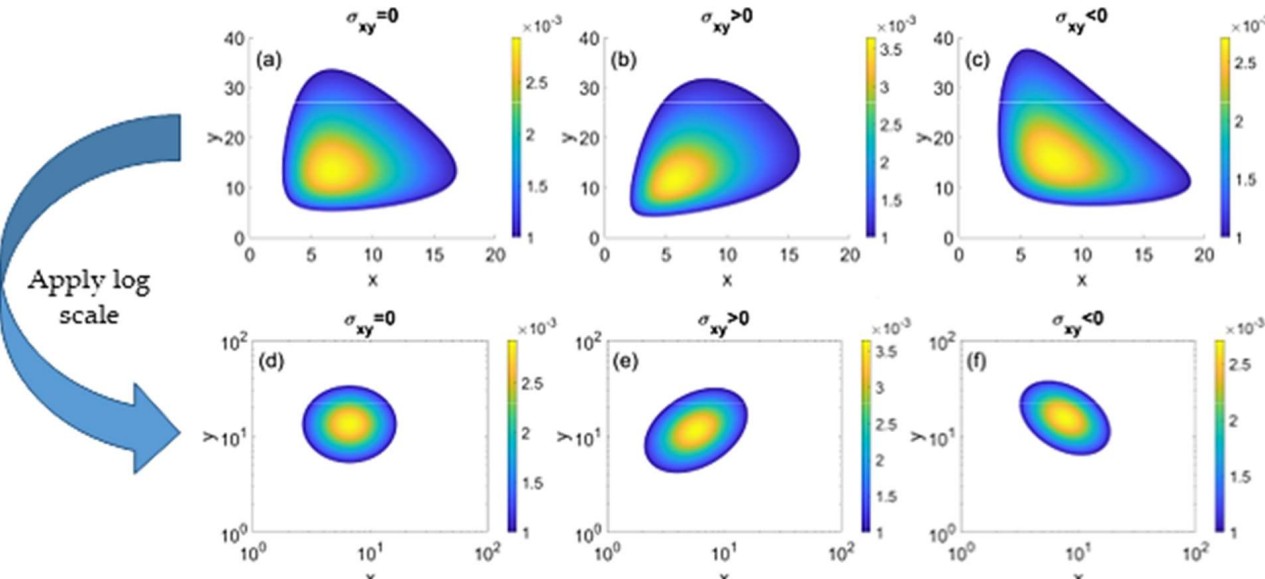

**Figure 1.** Effect of covariance on the lognormal distribution. (**a**,**d**) Lognormal distribution when the variables are uncorrelated. (**b**,**e**) Lognormal distribution when the variables have positive correlation. (**c**,**f**) Lognormal distribution when the variables have negative correlation. Plotting with a logarithmic scale as in (**d**–**f**) allows for visually interpreting the lognormal distribution as if it were a normal distribution.

We see from Figure 1a–c that interpreting a lognormal distribution is not as familiar as visually interpreting a normal distribution. For example, it is more difficult to immediately deduce if a correlation is positive, negative or zero when viewing only one of Figure 1a–c. However, if the lognormal distribution is plotted with a logarithmic scale on the axes such as in Figure 1d–f, then one identifies the visual properties of a normal distribution. For example, on the log scale, a vanishing correlation leads to a symmetrical distribution, a positive correlation tilts the distribution to move from the bottom-left to the top-right, whereas a negative correlation leads to the distribution moving from the top-left to the bottom-right.

Now that we have introduced the fundamental mathematical tools of analyzing multi-dimensional PSDs, we next discuss the details of the mathematical framework used to describe these particles.

### 3.2. Framework for Describing Multi-Dimensional Particle Property Distributions

The properties of particles depend on the distributions of size, shape, composition, surface and structure. As the PSD is the key parameter for describing a particle ensemble, here we focus on the mathematical description of multi-dimensional PSDs. It should be noticed that the framework for describing PSDs, presented in this section, can be transferred to other distributed particle properties, for example, surface, volume, band gap and composition. The general definition of a multi-dimensional PSD $q_r(\vec{x})$ is defined by Equation (8) [38–40].

$$q_r\left(\vec{x}\right) = \frac{amount\ of\ kind\ r\ in\ interval\ d\vec{x}}{interval\ size\ d\vec{x} \cdot total\ amount\ of\ kind\ r} \tag{8}$$

The index $r$ in Equation (8) defines the weighting of the PSD (0: number; 2: surface; 3: volume) and is governed by the physical principle of the particle size measurement method. The parameter $\vec{x}$ in Equation (8) is the particle property vector and describes the set of quantities required to characterize a particle completely. Only for spherical particles can the PSD be described by the distribution of a single size parameter, the particle diameter $x$. That means for spherical particles, $\vec{x} = x$ and the PSD is completely described by a one-dimensional density distribution of the particle diameter $q_r(x)$. For a one-dimensional PSD, it is well known that a $k$-weighted PSD can be converted into a $r$-weighted PSD via the moment method, Equation (9) [53–55].

$$q_r(x) = \frac{x^{r-k}q_k(x)}{M_{r-k,k}} = \frac{x^{r-k}q_k(x)}{\int_{xmin}^{xmax} x^{r-k}q_k(x)dx} \tag{9}$$

The extension of Equation (9) to multi-dimensional PSDs results in Equation (10) [38–40]

$$q_r\left(\vec{x}\right) = \frac{\kappa\left(\vec{x}\right)q_k\left(\vec{x}\right)}{M_{r,k}} = \frac{\kappa\left(\vec{x}\right)q_k\left(\vec{x}\right)}{\int_{\vec{x}\epsilon R^n} \kappa\left(\vec{x}\right)q_k\left(\vec{x}\right)d\vec{x}} \tag{10}$$

A $k$-weighted multi-dimensional PSD is converted into a $r$-weighted one through the pointwise multiplication of the $k$-weighted PSD with a weighting function $\kappa$ and dividing by the generalized moment $M_{r,k}$. The weighting function $\kappa$ depends on the property vector $\vec{x}$.

In the following, Equation (10) is applied for converting a number-weighted two-dimensional PSD of cylinders into surface- and volume-weighted ones. In this case, the property vector $\vec{x}$ depends on the length $l$ and the diameter $d$ of the cylinder according to Equation (11).

$$\vec{x} = (d,\ l)^T \tag{11}$$

The weighting function $\kappa$ in Equation (10) is replaced by functions for the surface $S(d,l)$ and volume $V(d,l)$ of the cylinders, Equations (12) and (13).

$$S(d,l) = \frac{1}{2}\pi d^2 + \pi dl, \tag{12}$$

$$V(d,l) = \frac{\pi}{4}d^2 l. \tag{13}$$

If the PSD $q_k\left(\vec{x}\right)$ follows the lognormal distribution in Equation (6) with parameters $\mu_k$ and $\Sigma_k$, the weighting function is of the form

$$\kappa\left(\vec{x}\right) = C\prod_{i=1}^{N} x_i^{b_i}, \tag{14}$$

where $C$ is a constant, $x_i$ is the components of $\vec{x}$, and $\vec{b} = (b_i)^{\mathrm{T}}$ is real numbers such that each component of $\mu_k + \Sigma_k \vec{b}$ is positive, then the converted PSD $q_r\left(\vec{x}\right)$ also follows a lognormal distribution with parameters $\mu_r = \mu_k + \Sigma_k \vec{b}$ and $\Sigma_r = \Sigma_k$. Hence, the conversion is very simple. A derivation of this result is found in the Supplementary Materials.

As an example, if $q_0$ is a number-based lognormal distribution with parameters $\mu_0$ and $\Sigma_0$, then the volume-weighted PSD of cylindrical particles is computed by using Equation (13) as the weighting function in Equation (10). In this case, Equation (13) satisfies the general form of Equation (14) with $\vec{b} = (2, 1)^{\mathrm{T}}$, which means that we immediately know that $q_3\left(\vec{x}\right)$ is lognormal with the following parameters:

$$\mu_3 = \mu_0 + \Sigma_0 \begin{bmatrix} 2 \\ 1 \end{bmatrix}, \quad \Sigma_3 = \Sigma_0. \tag{15}$$

While the mean size is shifted to larger values, in particular, the covariance is conserved. However, conversion from $q_0\left(\vec{x}\right)$ to the surface-weighted distribution $q_2\left(\vec{x}\right)$ with Equation (12) does not follow the pattern in Equation (14). Instead, we can decompose Equation (10) into the sum of two terms:

$$q_2\left(\vec{x}\right) = \frac{1}{M_{2,0}} \left( \frac{1}{2}\pi d^2 q_0\left(\vec{x}\right) + \pi d l q_0\left(\vec{x}\right) \right). \tag{16}$$

With some algebraic manipulations, we can show that Equation (16) can be expressed as the weighted sum of two lognormal distributions

$$q_2\left(\vec{x}\right) = w_1 \mathrm{LN}\left(\mu_0 + \Sigma_0 \begin{bmatrix} 2 \\ 0 \end{bmatrix}, \Sigma_0\right) + w_2 \mathrm{LN}\left(\mu_0 + \Sigma_0 \begin{bmatrix} 1 \\ 1 \end{bmatrix}, \Sigma_0\right), \tag{17}$$

where $\mathrm{LN}(\mu, \Sigma)$ indicates the parametrized lognormal shown in Equation (3), and the weights $w_1, w_2$ can be computed with the following equations:

$$C_1 = \exp\left[ \frac{1}{2} \begin{bmatrix} 2 & 0 \end{bmatrix} \left( \Sigma \begin{bmatrix} 2 \\ 0 \end{bmatrix} - 2\mu_0 \right) \right], \tag{18}$$

$$C_2 = \exp\left[ \frac{1}{2} \begin{bmatrix} 1 & 1 \end{bmatrix} \left( \Sigma \begin{bmatrix} 1 \\ 1 \end{bmatrix} - 2\mu_0 \right) \right], \tag{19}$$

$$w_1 = \frac{1/2\pi C_1}{1/2\pi C_1 + \pi C_2}, \tag{20}$$

$$w_2 = \frac{\pi C_2}{1/2\pi C_1 + \pi C_2} \tag{21}$$

We again provide this detailed derivation in the Supplementary Materials.

In Figure 2, the number-, surface- and volume-weighted two-dimensional lognormal PSDs of the cylinders are shown. The stars in Figure 2 represent the mode values of the PSDs, and the lines are the boundaries containing 95% of the distribution to demonstrate the shape of the PSDs.

According to Figure 2, the conversion from $q_0$ to $q_3$ shifts the two-dimensional PSD and its mode (maximum) value significantly to larger values for the diameter d and the length l. Moreover, the dispersity of lognormal distributions has a multiplicative relationship between the mean value and covariance, so having $\mu_0 < \mu_3$ and $\Sigma_0 = \Sigma_3$ results in $q_3$ being more disperse. This example demonstrates the importance of the weighting for comparing PSDs of different samples measured with different methods.

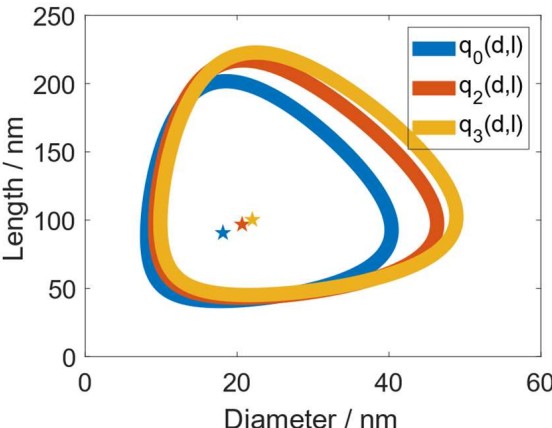

**Figure 2.** Comparison of a number- ($q_0$), a surface- ($q_2$) and a volume-weighted ($q_3$) two-dimensional PSD of nanocylinders. The stars represent the mode values (maximum values for $q_0$, $q_2$ and $q_3$, respectively), and the lines indicate regions enclosing 95% of the probability mass.

For a reduction in the amount of data, arbitrary *r*-weighted one-dimensional property functions can be calculated from two-dimensional PSDs by defining a linked variable. For rod- or plate-like particles, the aspect ratio $\nu$, which is defined as the length/diameter ratio, can be used as a linked variable [40]. A one-dimensional distribution of $\nu$ can be calculated by Equation (22) [39,40].

$$q(\nu) = \frac{\mathrm{d}}{\mathrm{d}\nu} \int_0^\infty \int_0^{\mathrm{d}\nu} q_0(d,l) \mathrm{d}l \; \mathrm{d}d \tag{22}$$

The application of Equation (22) can be interpreted as the calculation of the cumulative distribution of the aspect ratio $\nu$ with subsequent derivation to convert it into a density distribution [39,40]. In Figure 3, the application of this procedure is demonstrated to be exemplary for gold nanorods.

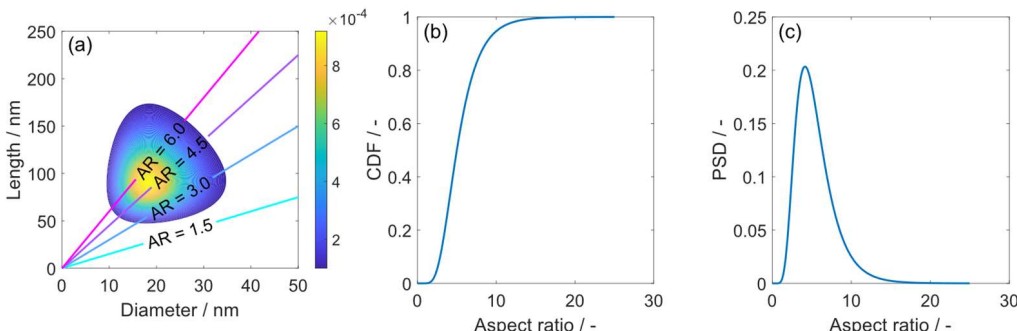

**Figure 3.** Calculation of the one-dimensional aspect ratio (AR) distribution from a two-dimensional PSD of nanocylinders: (**a**) two-dimensional lognormal PSD of cylinders, where the lines indicate different aspect ratios; (**b**) cumulative distribution of the aspect ratio distribution calculated via the integration of the area based on the aspect ratio contours; (**c**) density distribution of the aspect ratio obtained via the derivation of the cumulative distribution.

Another linked variable that is often interesting is the specific surface area *SSA*, which is the ratio of the surface area to volume (or mass). For the cylindrical gold nanorods, this is computed by

$$SSA = \frac{\pi d l + \frac{1}{2}\pi d^2}{\frac{\pi}{4}d^2 l} = \frac{4}{d} + \frac{2}{l} \tag{23}$$

The cumulative distribution function for the specific surface area $\rho(SSA)$ is computed by integrating over the region $(d,l)$, such that

$$\frac{4}{d} + \frac{2}{l} \leq SSA \tag{24}$$

This integration is more complicated than the aspect ratio distribution. Here, we isolate the variable $l$ algebraically by observing

$$\frac{2}{l} \leq \frac{SSA \cdot d - 4}{4}. \tag{25}$$

If $SAA \cdot d - 4 < 0$, then we can rearrange Equation (25) to see

$$l \leq \frac{-2d}{4 - SSA \cdot d} < 0 \tag{26}$$

which is not physical and, therefore, does not need to be considered. When $SAA \cdot d - 4 > 0$, i.e., $d > \frac{4}{S}$, we find that

$$l \geq \frac{2d}{SSA \cdot d - 4} \tag{27}$$

which is physically reasonable. Then, the one-dimensional distribution of the specific surface area can be computed with

$$\rho(SSA) = \frac{\mathrm{d}}{\mathrm{d}SSA} \int_{4/SSA}^{\infty} \int_{2d/(SSA \cdot d - 4)}^{\infty} q_0(d, l) \mathrm{d}l \, \mathrm{d}d \tag{28}$$

Applying this procedure to the example gold nanorods results in the distribution shown in Figure 4.

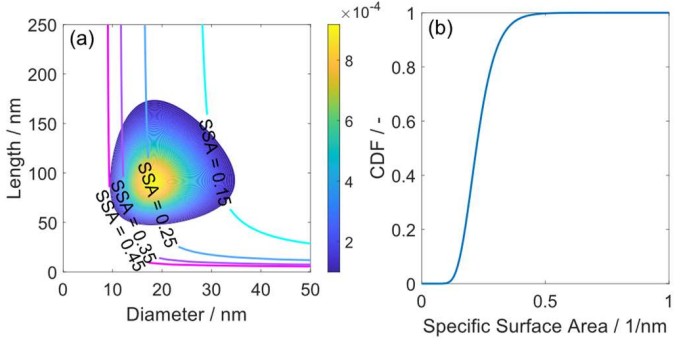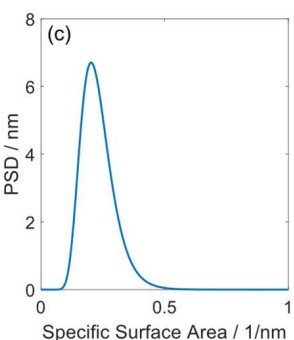

**Figure 4.** Calculation of the one-dimensional specific surface area (*SSA*) distribution in units $1/\mathrm{nm}$ from a two-dimensional PSD of nanocylinders: (**a**) two-dimensional lognormal PSD of cylinders, where the lines indicate different specific surface areas; (**b**) cumulative distribution of the specific surface area distribution calculated via the integration of the area based on the specific surface area contours; (**c**) density distribution of the specific surface area obtained via the derivation of the cumulative distribution.

### 3.3. Measuring Two-Dimensional Nanoparticle Distributions

#### 3.3.1. Overview

Particle systems are intrinsically multi-dimensional. Due to the underlying enormous complexity, mostly one-dimensional size distributions were analyzed in the past. The effective diameter approach was widely applied by assigning the properties of the real and often highly complex particle to an effective diameter of a sphere with the same physical properties as the non-spherical particle under consideration. This limitation stems from the difficulty of measuring multi-dimensional property distributions reliably and efficiently.

Continuous shape characterization using commercial video imaging systems is state-of-the-art for particles larger than roughly 10 microns. The particles must be well separated in a continuous flow for proper imaging. The recent advent of advanced characterization methodologies beyond size now also enables a comprehensive description of complex

nanoparticle systems, at least in two dimensions. Of course, SEM, TEM or AFM imaging is possible, if proper sample preparation is guaranteed and sufficient particles are counted for a statistically reliable number of particles. A recently published review on multi-dimensional particle characterization describes the current state of the art [39].

A particularly useful technique is analytical ultracentrifugation (AUC). AUC is a very old technique, for which the Nobel Prize was awarded in 1926 to Svedberg. Due to the high rotor revolutions of up to 60,000 rpm, nanoparticles from below 1 nm up to almost 1 μm can be analyzed. The technique has been routinely applied in biolabs for protein analysis since decades, whereas only a few labs widened to the scope to polymers and colloids. One of the pioneers was Helmut Cölfen, who recently passed away much too early [56,57]. The development of the new optical multi-wavelength absorption and emission sensors allows us to now routinely measure two-dimensional nanoparticle ensembles with unprecedented accuracy, reproducibility and statistically reliability. Separation in a centrifugal field via the settling particles is combined with their advanced optical characterization. The latter is particularly useful for particle systems with size-, shape- and composition-dependent optical properties. Applications include the following:

- Size and shape characterization of noble metal nanorods with specific plasmonic resonances or 2D materials such as graphene (oxide) [58] or molybdenum disulfide;
- Size and density characterization of alloys [59] or core–shell systems such as nanoparticles covered by adsorbed stabilizing layers [60];
- Size and bandgap characterization of semiconducting nanoparticles, see, for instance, [56].

For the time being, the application of AUC techniques is restricted to a limited number of advanced labs in academia and industry. Therefore, the question arises to what extent techniques for one-dimensional characterization can be used to deduce 2D PSDs. Any 1D PSD can be derived from a 2D PSD via the approach discussed above. However, the opposite, i.e., the reconstruction of 2D PSDs from 1D PSDs, is an ill-posed problem and will, in general, not lead to a unique 2D PSD. In [61], it is shown how a two-dimensional property space can be constructed via the combination of two one-dimensional measurements. The presented reconstruction method allows the estimation of the bivariate distribution of the lengths and diameters of gold nanorods using solely their particle mass and extinction-weighted sedimentation coefficient distributions.

### 3.3.2. Molybdenum Disulfide Nanosheets

Molybdenum disulfide ($MoS_2$) is a semiconductor material with size-dependent optical properties [62,63]. It exhibits excitonic transitions (A, B, C and D exciton) in the visible part of the electromagnetic spectrum, which are shifting with the lateral diameter and the thickness of the nanosheets (see Figure 5a) [63].

C. Backes et al. developed a methodology for extracting mean values of the lateral diameter $L$ (in μm) and the number of layers per particle $N$ from UV/Vis-spectra via the following correlations, shown in Equations (29) and (30) [63]:

$$L = \frac{3.5\ \mu m \frac{E_B}{E_{345nm}} - 0.14\ \mu m}{11.5 - \frac{E_B}{E_{345nm}}} \tag{29}$$

$$N = 2.3 \cdot 10^{36} e^{-\frac{54{,}888nm}{\lambda_A}} \tag{30}$$

$E_B$ and $E_{345nm}$ in Equation (29) are the extinction values at the B exciton wavelength and 345 nm, respectively, and $\lambda_A$ in Equation (30) is the wavelength of the A exciton in nm [63]. Equations (29) and (30) are valid for $MoS_2$-nanosheets with lateral diameters from 70 to 350 nm and fewer than 10 layers [63].

Based on the very convenient spectroscopic access to mean values for $L$ and $N$ by Equations (29) and (30), the question of how to measure distributions for $L$ and $N$ in suspension arose. To make a rough estimate for distributions, we centrifuged our $MoS_2$-feed sample "fraction-300 nm" to different sphere-equivalent cut sizes, as described in Section 2.2,

and evaluated the UV/Vis-spectrum of each size fraction using Equations (29) and (30). In Figure 5c,d, the obtained values for $L$ and $N$ are presented as a function of the sphere-equivalent cut size. Thinner and smaller nanosheets are obtained if the sphere-equivalent cut size becomes smaller (see Figure 5c,d).

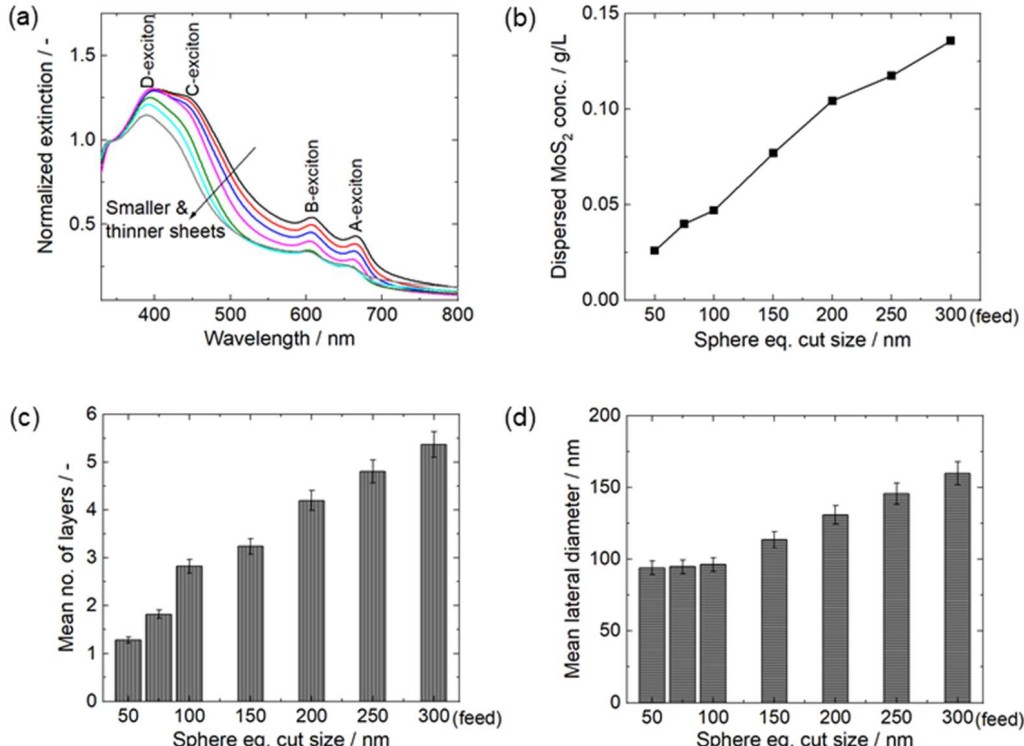

**Figure 5.** Normalized UV/Vis-spectra of $MoS_2$-nanosheets in water as a function of the size of the nanosheets, the different colored lines indicate different nanosheet sizes with lateral size decreasing from 160 nm (black line) to 94 nm (gray line) (**a**), dispersed $MoS_2$ concentration (**b**), number of layers N (**c**) and lateral diameter L (**d**) as a function of the sphere-equivalent cut size.

Moreover, the dispersed $MoS_2$-concentration in each size fraction is accessible from the extinction value at 345 nm via the Lambert–Beer law (extinction coefficient: 69 L/g cm) [63]. According to Figure 5b, the concentration of the sample increases if the centrifugation cut size becomes coarser.

The distributions of the lateral diameter $L$ and the number of layers $N$ were constructed from the values for $L$, $N$ and the $MoS_2$-concentration of each size fraction as follows: The $MoS_2$-concentration of each fraction was normalized to the concentration value for the feed material ("fraction-300 nm", 0.14 g/L). For each size fraction, the normalized $MoS_2$-concentration was plotted vs. the corresponding values for $L$ and $N$, respectively. The resulting plots are shown in Figure 6 and can be interpreted as estimates for the extinction-weighted cumulative distributions of the lateral diameter $L$ and the number of layers $N$ of the "fraction-300 nm" sample. As the extinction of the samples at 345 nm is almost independent of the size of the $MoS_2$-nanosheets [63] and, therefore, proportional to the mass concentration of $MoS_2$, the extinction-weighted distributions correspond to mass-weighted distributions ($Q_3$). The distribution of the number of layers $N$, shown in Figure 6a, is almost linear, and a remarkable amount (about 30%) of very thin $MoS_2$-nanosheets with $N \leq 2$ is visible. The distribution of the lateral diameter $L$, shown in Figure 6b, is well described by a sigmoidal Weibull-function in Equation (31), which is shown by the curve in Figure 6b.

$$Q = Q_{max}\left[1 - \exp\left\{-[k(L - L_c)]^f\right\}\right] \tag{31}$$

where $Q_{max}$ is the maximum value of the cumulative distribution, $L_c$ is the location parameter (the threshold value), $k$ is the scale parameter and $f$ is the shape parameter.

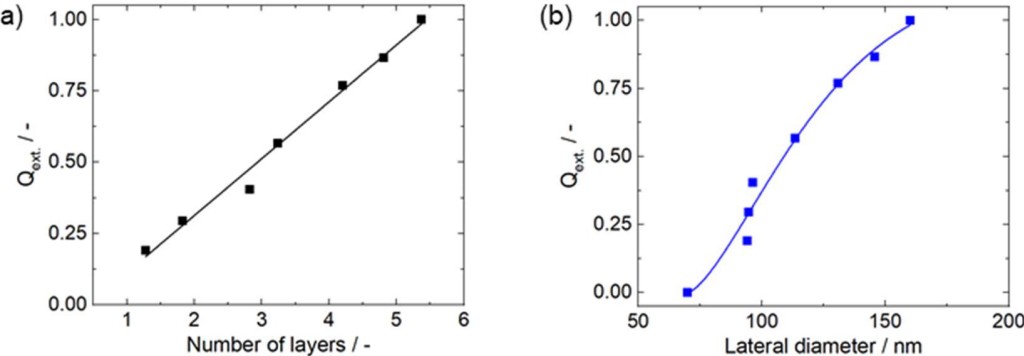

**Figure 6.** Estimates for the extinction-weighted cumulative distribution of the number of layers $N$ (**a**) and the lateral diameter $L$ (**b**) for the MoS$_2$ sample "fraction 300 nm". The datapoints in both diagrams are the measured values, and the lines are fitting functions (linear for $N$ and Equation (31) for $L$ with $Q_{max} = 1$, $L_c = 70$ nm, $k = 0.018$ and $f = 1.47$).

It should be noted that the distributions of $L$ and $N$ can be measured more conveniently and more precisely via MWL-AUC. A detailed MWL-AUC study on different MoS$_2$ size fractions classified via preparative centrifugation including separation efficiencies for the different platelet dimensions is currently in progress. The powerful in-suspension method MWL-AUC for measuring two-dimensional PSDs and its application to gold nanorods is presented in the next section.

### 3.3.3. Gold Nanorods

The length and diameter distributions of commercial gold nanorods were measured via MWL-AUC. This powerful method measures the distribution of the sedimentation coefficient $s$ (the sedimentation rate normalized to the centrifugal acceleration) in combination with the diffusional and optical properties (UV/Vis/NIR-spectrum) of a particle ensemble in suspension and is considered to be the gold standard for particle characterization due to its high precision [47,48,64,65]. The combined information obtained from MWL-AUC allows for the determination of multi-dimensional particle property distributions. The sedimentation coefficient $s$, which is defined by Equation (32), depends on the size (via the mass $m$ and the volume-equivalent diameter $x_V$), the shape (via the frictional ratio $f/f_0$, $f/f_0 = 1$ for spheres and larger than 1 for non-spherical particles) and the composition of the particle (via the particle density $\rho_{particle}$), as well as on the density $\rho_{solvent}$ and viscosity $\eta$ of the solvent.

$$s = \frac{m\left(1 - \frac{\rho_{solvent}}{\rho_{particle}}\right)}{3\pi\eta x_V \frac{f}{f_0}} \tag{32}$$

The measured distribution of the sedimentation coefficient can be subdivided into different intervals (Figure 7a), and for each interval, the extinction spectrum can be extracted using AUC-software (https://www.aucegypt.edu/digital-transformation/software-solutions, accessed on 6 February 2024) (Figure 7b). Gold nanorods exhibit surface plasmon resonance and, therefore, their spectra are strongly dependent on their size and shape, in particular on their aspect ratio. As the surface plasmon resonance of gold nanorods can be described quantitatively using physical models, two-dimensional size distributions are accessible from the combined sedimentation data and optical properties. The UV/Vis-spectra of the gold nanorods were modeled via the classical Gans-theory, which approximates the gold nanorods as spheroids [66–68]. Additionally, an orientationally averaged finite element method (FEM) and an analytical longitudinal polarization model (LP model) were used for modeling the spectra [69]. The FEM and the LP methods approximate the gold nanorods

as cylinders with hemispherical endcaps. The latter one takes only the longitudinal surface plasmon resonance into account, because it is much stronger than the transversal surface plasmon resonance [66]. The Mie–Gans model was used to fit the experimental spectra under the constraint of the sedimentation coefficient. The deconvolution of the spectrum yields the distribution of the aspect ratio of the gold nanorods, but it does not determine the length and diameter values unambiguously, as many length–diameter combinations result in the same aspect ratio. Rather, as the volume of the particles is fixed and accessible from the sedimentation coefficient, belonging to the extinction spectrum, the values for length and diameter of the gold nanorods can be univocally determined.

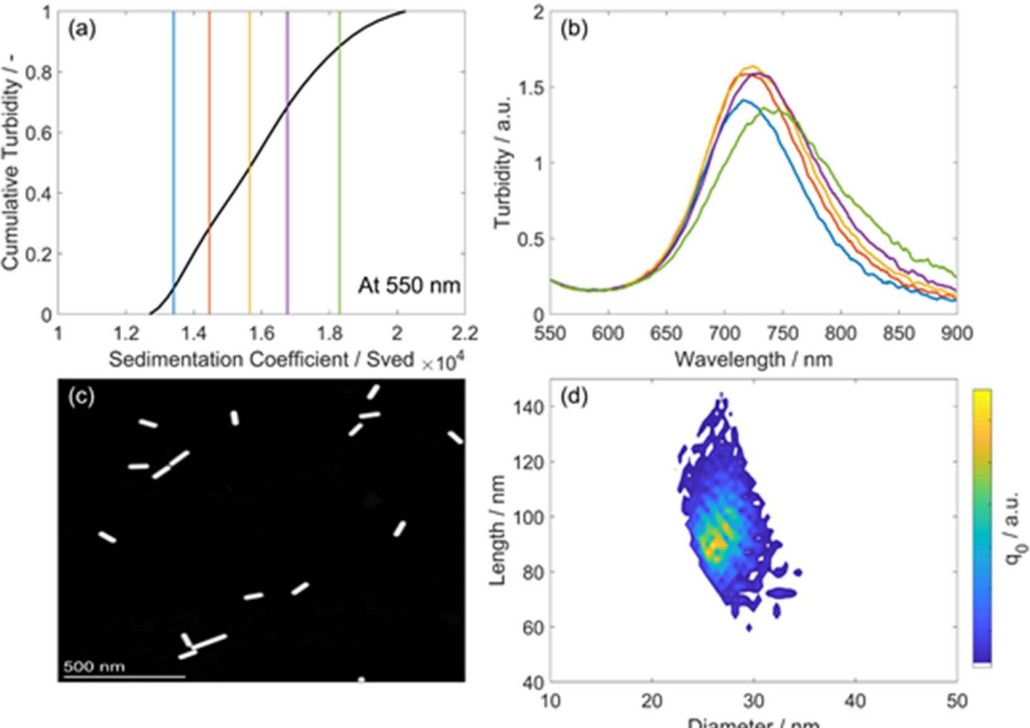

**Figure 7.** Sedimentation coefficient distribution of a gold nanorod sample with marked sedimentation coefficients (**a**). Extracted extinction spectra for each marked sedimentation coefficient (**b**). The color of the line/graph in (**a**,**b**) marks the sedimentation coefficient-spectrum couples. Typical TEM-micrograph of the gold nanorod sample (**c**). Two-dimensional length and diameter distribution of the gold nanorod sample measured via the optical back coupling MWL-AUC method (**d**). The color code in (**d**) indicates number density $q_0$ of a certain length-diameter combination with $q_0$ increasing from blue via green to orange.

This so-called optical back coupling method for measuring two-dimensional size distributions of gold nanorods was validated via the statistical evaluation of TEM micrographs [69]. A typical TEM micrograph for the herein selected gold nanorod sample is shown in Figure 7c. The two-dimensional size distribution (length and diameter) measured using MWL-AUC via the optical back coupling method is depicted in Figure 7d.

Spherical gold nanoparticles are a typical side product of gold nanorod synthesis. As gold nanorods and spherical gold nanoparticles exhibit distinct extinction-spectra, the MWL-AUC optical back coupling method allows the quantitative determination of the mass fractions of rods and spheres [42]. In addition, the diameter distribution of the spheres fraction and the length and diameter distribution of the rod fraction are accessible [42]. In conclusion, this method allows the fast and highly accurate comprehensive characterization of products from nanoparticle syntheses. Recently, the method was extended to five-sided double pyramids [41] and the size and composition of Au–Ag alloy nanoparticles [59].

3.3.4. Fitting of Measured Data to Multi-Dimensional Lognormal Distributions

The analysis methodologies described in Section 3.2 become much simpler when the data are reasonably described via a normal or lognormal distribution. Thus, it stands to reason to have strategies to check if experimental data closely follow a normal or lognormal distribution. When testing if data are lognormally distributed, the best strategy is to log-transform the data and perform analyses that are relevant to the normal distribution. Hence, we will only discuss finding parameters of a normal distribution and testing if data are normally distributed and note that the exact procedure can be applied for lognormal distributions, with only the additional step of log-transforming the data beforehand.

There are two primary types of data that one can receive for the PSD. First are counts of individual particles such as in a SEM or TEM measurement. Second are ensemble techniques, which provide estimates of the probability density for particular sizes.

Let us first address the case of counting data such as those of from TEM. Here, the parameter estimation is fortunately quite simple. The parameter for the mean value $\mu$ is best estimated with the sample mean

$$\mu \approx \bar{x} = \frac{1}{N}\sum_{i=1}^{N}\vec{x_i},$$ 
(33)

where $\vec{x_i}$ is the measured particles. The covariance matrix $\Sigma$ is best described by

$$\Sigma \approx \frac{1}{N-1}\sum_{i=1}^{N}\left(\vec{x_i}-\bar{x}\right)^{T}\left(\vec{x_i}-\bar{x}\right).$$ 
(34)

With TEM data, the combination of Equations (33) and (34) is the maximum likelihood estimator [70] of the parameters of a multi-dimensional (log)normal distribution.

After performing parameter estimation with Equations (33) and (34), the next task is to determine how well the (log)normal distribution model describes the data. With multi-dimensional distributions, this task is slightly more complicated than that with one-dimensional particles. At first, graphical assessment can be performed with a Q-Q plot. Statistical theory regarding normal distributions shows that the squared Mahalanobis distance $d_M^2$ has the following property [71]:

$$\left(\vec{x}-\mu\right)^{T}\Sigma^{-1}\left(\vec{x}-\mu\right) = d_M^2 \sim \chi_N^2$$ 
(35)

where $\chi_N^2$ is the chi-squared distribution with $N$ degrees of freedom, where $N$ is the number of dimensions of the particle. Therefore, with our estimations of $\mu$, $\Sigma$ and data $\vec{x_i}$, values of $d_{Mi}^2$ can be computed for each data point, and then empirical quantiles can be computed [72]. The theoretical quantiles of the chi-squared distribution can then be computed with the inverse CDF and a scatter plot constructed by pairing the $d_M^2$ value of a particular quantile with the theoretical chi-squared value of the same quantile; this is the so-called Q-Q plot. If the Q-Q plot indicates a linear relationship, then this is evidence that the data are normally distributed. In Figure 8, we provide an example of this graphical analysis applied to the TEM data of gold nanorods presented in [42]. Of course, the Q-Q plots do not show perfect alignment, but there is still information to be gained by interpreting the results. For example, in Figure 8b, we see that when $d_M^2 < 5$, there appears to be a linear trend, but for $d_M^2 > 5$, the trend is still linear, but the slope is shallower. This behavior provides a hint that perhaps the sum of two different normal distributions would be a better model. Similarly, in Figure 8d, we see what appear to be two, separate, linear trends where the change in slope occurs at $d_M^2 \approx 6.5$, again indicating that two lognormal distributions may be a more appropriate model.

There are also more rigorous, quantitative assessments that can be used, but their discussion is out of the scope of this work; see, for example, the review for details [73]. Testing the assumption of a normal distribution is a well-studied field in statistics, and there are a number of software packages that perform this analysis well. In particular,

the R programming language has among the most mature software implementations, for instance, the MVN library [74].

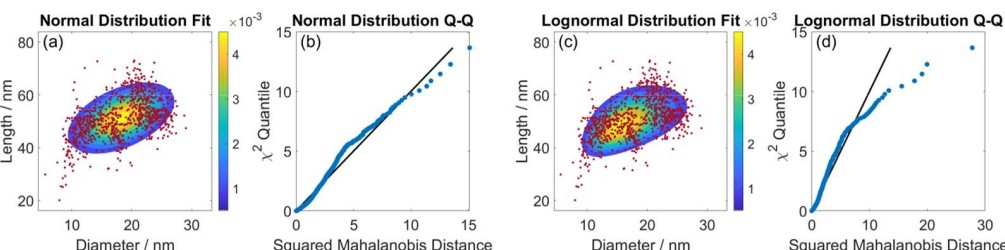

**Figure 8.** Graphical assessment of how well normal and lognormal distributions fit example data of gold nanorods measured with TEM [42]. (**a**,**c**) show the best fit distributions as contours with the observed data overlayed with a scatter plot. (**b**,**d**) show the empirical versus theoretical quantiles, as computed using Equation (35). If data are well described by the normal or lognormal distribution, then the Q-Q plot will show the blue points following the black line closely.

When the data are estimates of the probability density of the PSD, a more involved optimization routine is necessary to estimate the parameters $\mu$ and $\Sigma$. A necessity, then, is to find a robust way to quantify the difference between two probability distributions. One way to accomplish this is to use the Kullback–Leibler divergence (KL-divergence) [75] defined as

$$D_{KL}(P\|Q) = \int_{-\infty}^{\infty} p\left(\vec{x}\right) \ln\left(\frac{p\left(\vec{x}\right)}{q\left(\vec{x}\right)}\right) d\vec{x}, \tag{36}$$

where $P, Q$ are the probability distributions being compared, and $p\left(\vec{x}\right), q\left(\vec{x}\right)$ are their associated probability density functions. Here, the distribution $P$ represents the observed data, and the distribution $Q$ represents the theoretical distribution that we wish to use to model the data—in this case, a normal distribution. Equation (36) may not be as intuitive as functions used when comparing two curves, such as the mean squared error. This is because comparing the "distance" between two probability distributions is more abstract than comparing the distance between two curves. However, through a field of mathematics called Information Geometry, one can show that KL-divergence satisfies a generalized form of the Pythagorean theorem, and, therefore, Equation (36) may be conceptualized as the mean squared error applied to probability distributions [76].

Then, our task is to solve the following optimization problem:

$$\mu^*, \Sigma^* = \underset{\mu,\Sigma}{\operatorname{argmin}} D_{KL}(\text{Data}\|\text{Model}) = \underset{\mu,\Sigma}{\operatorname{argmin}} \int_{-\infty}^{\infty} q_{\text{data}}\left(\vec{x}\right) \ln\left(\frac{q_{\text{data}}\left(\vec{x}\right)}{q_{\text{theory}}\left(\vec{x}\right)}\right) d\vec{x}. \tag{37}$$

The optimization problem in Equation (37) can then be solved with numerical optimization techniques. The result of solving Equation (37) is the optimal parameters that define the (log)normal distribution that best fits the data. This procedure can be performed with any other parameterized distribution as well. One would simply need to update $q_{\text{theory}}\left(\vec{x}\right)$ and the parameters $\mu$, $\Sigma$ in Equation (37) to reflect the parameters and probability density function in the alternative model. If $q_{\text{data}}$ exactly matches $q_{\text{theory}}$, then the KL-divergence would be zero. Thus, multiple different parametrizations can be tested for how well they fit the data, and the model whose KL-divergence is closest to zero would correspond to the best fit.

The KL-divergence is difficult to interpret in terms of the computed number. To better understand how good a fit the procedure in Equation (37) provides, one can calculate

the relative amount of mass within a region for the raw data and the fit distribution. For example, for a two-dimensional particle, one can compute

$$M_{\text{data}} = \int_{x_1^{\min}}^{x_1^{\max}} \int_{x_2^{\min}}^{x_2^{\max}} \text{Volume}(x_1, x_2) \times q_{\text{data}}(x_1, x_2)\mathrm{d}x_2\mathrm{d}x_1, \tag{38}$$

$$M_{\text{theory}} = \int_{x_1^{\min}}^{x_1^{\max}} \int_{x_2^{\min}}^{x_2^{\max}} \text{Volume}(x_1, x_2) \times q_{\text{theory}}(x_1, x_2)\mathrm{d}x_2\mathrm{d}x_1, \tag{39}$$

$$\text{Relative mass difference} = \frac{M_{\text{data}} - M_{\text{theory}}}{M_{\text{theory}}} \times 100\%, \tag{40}$$

in order to calculate a percentage difference in the particle mass contained within the rectangle spanning $x_1^{\min} \leq x_1 \leq x_1^{\max}$ and $x_2^{\min} \leq x_2 \leq x_2^{\max}$. A large number of small rectangles would provide a fine-grained analysis of where a fewer or greater number of particles are located in the fit (log)normal distribution as compared to the data; a small number of large rectangles would provide information on how well the (log)normal distribution describes the data.

In some cases, a measured PSD displays multimodal behavior (i.e., two or more defined peaks). When this occurs, fitting a single (log)normal distribution is no longer the best approach. An alternative is to suggest a mixture model where two or more (log)normal distributions are assumed to compose the total PSD. Then, the parameters of the mixture model would be estimated numerically to determine the underlying distributions that make up the PSD. This parameter estimation turns out to be more difficult than what we have described thus far—mathematically and statistically speaking—and is, therefore, out of the scope of this work. However, we note that the so-called expectation maximization algorithm is often what is used to solve these sorts of problems [77].

### 3.4. Describing Particle Separation Processes—From the One-Dimensional to the Multi-Dimensional Case

In the simplest case, one separation process is applied to a one-dimensional particle ensemble with the PSD $q_{feed}(x)$. One separating force acts on the particles and drags them, depending on their size, to different positions in the separation device [53,55]. This leads to the classification of the particle ensemble in a coarse fraction with the relative amount $c$ and the PSD $q_{coarse}(x)$, as well as in a fine fraction with the relative amount $f$ and the PSD $q_{fine}(x)$. Between feed, coarse and fine fractions, the balances defined by Equations (41) and (42) have to be fulfilled:

$$c + f = 1 \tag{41}$$

$$q_{feed}(x) = c{\cdot}q_{coarse}(x) + f{\cdot}q_{fine}(x) \tag{42}$$

The one-dimensional separation process is quantitatively described via the separation efficiency curve $T(x)$, which is defined by Equation (43) and gives the probability of finding a particle with a size in the interval $dx$ in the coarse fraction.

$$T(x) = \frac{\text{amount of particles in the coarse in size interval } dx}{\text{amount of particles in the feed in size interval } dx} = c{\cdot}\frac{q_{coarse}(x)}{q_{feed}(x)} = 1 - \frac{f{\cdot}q_{fine}(x)}{q_{feed}(x)} \tag{43}$$

The particle size that is found in the coarse and fine fraction with the same probability is called cut size $x_{cut}$ with $T(x_{cut}) = 0.5$. If T(x) and the feed PSD $q_{feed}(x)$ are known, $c$, $q_{coarse}(x)$, $f$ and $q_{fine}(x)$ can be calculated by Equations (44)–(47), and, therefore, the outcome of the separation process can be predicted.

$$q_{coarse}(x) = \frac{1}{c} {\cdot} T(x) {\cdot} q_{feed}(x) \tag{44}$$

$$c = \int T(x) \cdot q_{feed}(x) dx \tag{45}$$

$$q_{fine}(x) = \frac{1}{f} \cdot (1 - T(x)) \cdot q_{feed}(x) \tag{46}$$

$$f = \int (1 - T(x)) \cdot q_{feed}(x) dx \tag{47}$$

We extended this traditional approach for one-dimensional separations to the multi-dimensional case [38–40]. In the general view, m separation processes are applied to a multi-dimensional particle ensemble with the PSD $q_{feed}(\vec{x})$. Each single separation step leads to the fractionation of the feed particle ensemble into a coarse and a fine fraction. The total number of fractions B depends on the separation techniques used. In the multi-dimensional case, the separation curve is a function of the property vector (in Equation (41), the particle size $x$ has to be replaced by the property vector $\vec{x}$). For each of the m separation processes, there is one multi-dimensional separation efficiency curve $T_i(\vec{x})$ (with $i = 1\ldots$m). All separations that classify particles into the fraction $j$ can be divided into two sets: $K_j$ ($i = 1\ldots$m) if $T_i(\vec{x})$ classifies the particles into the coarse and $P_j$ ($i = 1\ldots$m) if $T_i(\vec{x})$ classifies the particles into the fines. Using these sets, the PSD in fraction $j$ $q_j(\vec{x})$ (with $j = 1\ldots$B) can be calculated by Equation (48).

$$q_j\left(\vec{x}\right) = \frac{1}{\gamma_j} \prod_{K_j} T_i\left(\vec{x}\right) \cdot \prod_{P_j} \left(1 - T_i\left(\vec{x}\right)\right) \cdot q_{feed}\left(\vec{x}\right) \tag{48}$$

The parameter $\gamma_j$ in Equation (48) is the relative amount of particles in the fraction $j$ and can be calculated by Equation (49).

$$\gamma_j = \int \prod_{K_j} T_i\left(\vec{x}\right) \cdot \prod_{P_j} \left(1 - T_i\left(\vec{x}\right)\right) \cdot q_{feed}\left(\vec{x}\right) d\vec{x} \tag{49}$$

If the feed PSD $q_{feed}(\vec{x})$ and the separation efficiency curves for each separation $T_i(\vec{x})$ (with $i = 1\ldots$m) are known, the outcome of any multi-dimensional separation process can be predicted, as the relative amounts of particles $\gamma_j$ and the PSD of each fraction $q_j(\vec{x})$ can be reconstructed using Equations (48) and (49).

In the following, this general approach for describing multi-dimensional separation processes is applied to different examples of the fractionation of gold nanorods to make it more demonstrative. In the first example, an ensemble of gold nanorods with lognormal distributions of the length $l$ and the diameter $d$ is fractionated regarding the volume (for example, via centrifugation). In this case, one separating force acts on a two-dimensional particle ensemble and classifies it into two two-dimensionally distributed fractions (coarse and fines). A linear separation efficiency curve is assumed for the volume $T(V)$ (see Figure 9a). The black line in Figure 9a marks the cut volume $V_{cut}$ with $T(V_{cut}) = 0.5$, and the red and the green lines mark the lower and upper separation boundaries for the volume. Figure 9b shows the two-dimensional separation efficiency curve $T(l,d)$, which was constructed from $T(V)$, assuming cylinder geometry for the gold nanorods.

The application of the general Equations (48) and (49) to this separation task results in the following equations for calculating the relative amounts of coarse and fine fractions $c$ and $f$ and their PSDs:

$$q_{coarse}(l,d) = \frac{1}{c} \cdot T(l,d) \cdot q_{feed}(l,d), \tag{50}$$

$$c = \int T(l,d) \cdot q_{feed}(l,d) \, \mathrm{d}l \, \mathrm{d}d, \tag{51}$$

$$q_{fine}(l,d) = \frac{1}{f} \cdot (1 - T(l,d)) \cdot q_{feed}(l,d), \tag{52}$$

$$f = \int (1 - T(l,d)) \cdot q_{feed} \, dl \, dd. \tag{53}$$

The PSDs of the feed, the coarse and the fine fractions are shown in Figure 10. The cut size $T(V_{cut}) = 0.5$ (black line) goes through the mean feed particle size and splits the PSD in a coarse and fine fraction. The red lines mark the lower and upper separation boundaries. Figure 10b,c show that the separation is not ideally sharp as not each particle with $V < V_{cut}$ is sorted into the fine fraction and vice versa. Misplaced particles are visible as contribution to the PSD below the black line for the coarse and above the black line for the fines.

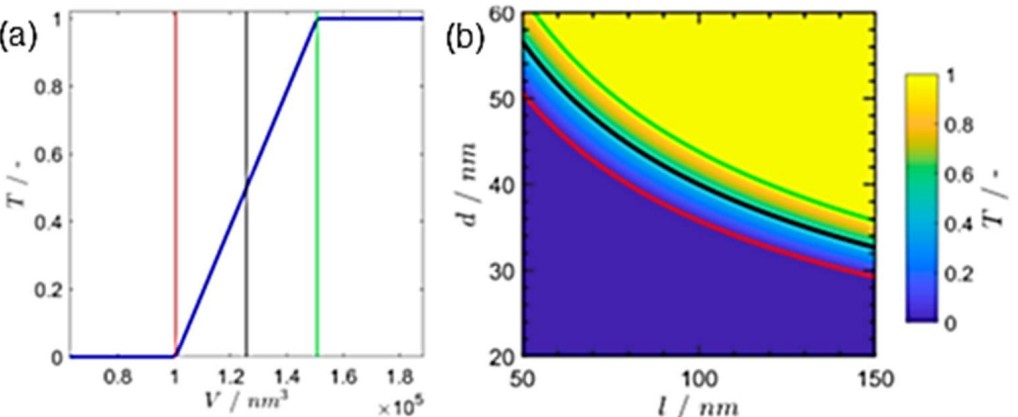

**Figure 9.** Separation efficiency curve for the volume of gold nanorods. (**a**) The red and green lines mark the lower and upper separation boundaries for the volume, and the black line indicates the cut volume $T(V_{cut}) = 0.5$. Two-dimensional separation efficiency curve $T(l,d)$ constructed from $T(V)$ approximating the gold nanorods as cylinders. (**b**) The lines mark isolines with same volume for the lower and upper boundary and the cut size. Reproduced with permission from [38], copyright 2023, MDPI.

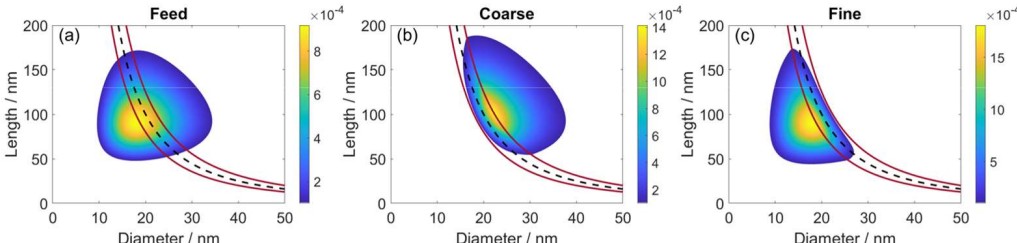

**Figure 10.** (**a**) Lognormal PSD of the gold nanorod feed sample; (**b**) PSD of the coarse fraction; (**c**) PSD of the fine fraction. The black line indicates the cut volume $T(V_{cut}) = 0.5$. The red lines mark isolines with same volume for the lower and upper separation boundaries.

In the next example, two orthogonal separations are applied to the gold nanorod ensemble [38]. The particles are classified regarding their sedimentation coefficient $s$ by the centrifugal force $F_s$ and regarding their electrophoretic mobility $\mu_M$ by the electrical field $F_{el}$ acting orthogonally to the centrifugal field. The sedimentation coefficient is calculated with Equation (32), and the electrophoretic mobility is computed with Equation (54)

$$\mu_M = \frac{A\rho_{ch}}{3\pi\eta x_V f / f_0}, \tag{54}$$

where $A$ is the surface area of the particle, $\rho_{ch}$ is the surface charge density and the denominator is the same as described in Equation (32).

The result of this combined separation process is four fractions, which are defined as follows:

- Fraction I: $s \geq s_{cut}$ and $\mu_M \geq \mu_{M,cut} \geq \rightarrow$ coarse with respect to $s$ and $\mu$;
- Fraction II: $s \geq s_{cut}$ and $\mu_M < \mu_{M,cut} \rightarrow$ coarse with respect to $s$ but fine with respect to $\mu$;
- Fraction III: $s < s_{cut}$ and $\mu_M \geq \mu_{M,cut} \rightarrow$ fine with respect to $s$ but coarse with respect to $\mu_M$;
- Fraction IV: $s < s_{cut}$ and $\mu_M < \mu_{M,cut} \rightarrow$ fine with respect to $s$ and $\mu_M$.

The multi-dimensional separation efficiency can be defined for each of these regions as follows:

- Fraction I: $T_I(s, \mu_M) = T_{Sed}(s) \times T_{Mobility}(\mu_M)$;
- Fraction II: $T_{II}(s, \mu_M) = T_{Sed}(s) \times \left(1 - T_{Mobility}(\mu_M)\right)$;
- Fraction III: $T_{III}(s, \mu_M) = (1 - T_{Sed}(s)) \times T_{Mobility}(\mu_M)$;
- Fraction IV: $T_{IV}(s, \mu_M) = (1 - T_{Sed}(s)) \times \left(1 - T_{Mobility}(\mu_M)\right)$.

For the separation efficiency curves $T_{Sed}(s)$ and $T_{Mobility}(\mu_M)$, linear functions between some pre-defined upper and lower bounds are again assumed.

Through Equations (48) and (49), the PSDs in each separation fraction I–IV can be computed. Often, the desired outcome of a separation process is to optimize the PSD area that is captured in one of the fractionation regions. Therefore, an optimization problem can be formulated based on Equation (49). As a demonstrative example, we formulate the following optimization problem.

Suppose our goal is for $Z\%$ of the feed to be separated into fraction II. Assume that the upper and lower boundaries of the separation efficiency curves described by $T_{Sed}(s)$ and $T_{Mobility}(\mu_M)$ can be experimentally modified independently of one another, but the width of each efficiency is restricted by some minimum values $s_{\text{width}}$ and $\mu_{M,\text{width}}$. Let the lower boundaries of each separation curve be denoted $s_{\text{lower}}$ and $\mu_{\text{lower}}$, respectively, and the upper boundaries be $s_{\text{upper}}$ and $\mu_{\text{upper}}$. Define

$$\theta = \begin{pmatrix} s_{\text{lower}} & s_{\text{upper}} & \mu_{M,\text{lower}} & \mu_{M,\text{upper}} \end{pmatrix}^{\text{T}} \tag{55}$$

as the optimization parameters. Then, we want to solve

$$\theta^* = \underset{\theta}{\operatorname{argmin}}(\gamma_{II} - Z)^2, \tag{56}$$

where $\gamma_{II}$ is the area of region II, as defined in Equation (49). Equation (56) is subject to the constraints

$$s_{\text{lower}} - s_{\text{upper}} \leq -s_{\text{width}}, \tag{57}$$

$$\mu_{M,\text{lower}} - \mu_{M,\text{upper}} \leq -\mu_{M,\text{width}}, \tag{58}$$

and perhaps upper and lower bounds on the parameters $\theta$. We set $Z = 0.25$ and suggest that the minimum attainable width is half an order of magnitude for both the sedimentation coefficient and electrophoretic mobility. Figure 11 shows the PSDs of the fractions I-IV after solving the constrained optimization problem defined by Equations (56)–(58). The fraction in region II is, indeed, found to be 25% after the optimization procedure is applied.

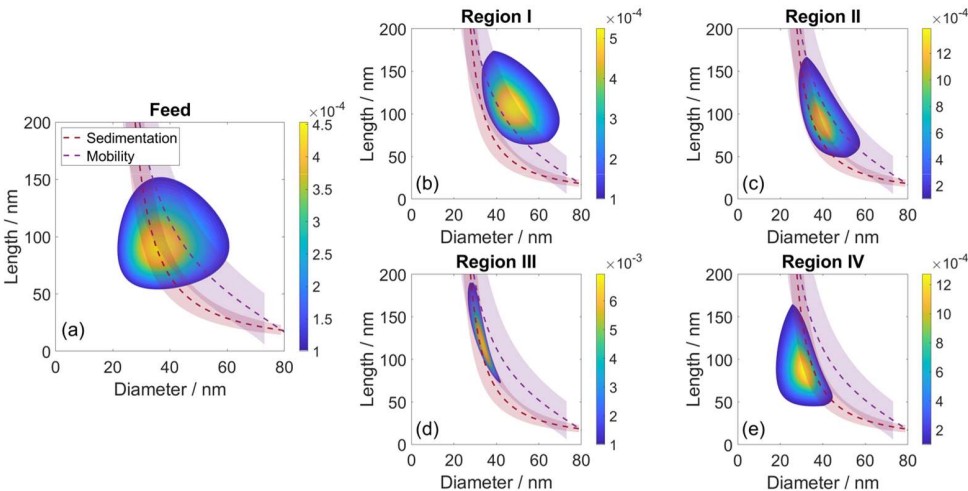

**Figure 11.** (**a**) The feed PSD with optimized cuts based on the sedimentation coefficient and electrophoretic mobility. The dashed line indicates the midpoint of the efficiency curve, and the shaded regions indicate the widths of the efficiency curve extending from the lower to the upper boundaries. (**b**) The coarse-coarse PSD after separation. (**c**) The coarse-fine PSD after separation. The separation efficiency curves were optimized such that this area was close to 25% of the original feed PSD, as calculated by Equation (49). Indeed, upon optimization, Equation (49), for this region, equals 25%. (**d**) The fine-coarse PSD after separation. (**e**) The fine-fine PSD after separation.

## 4. Discussion

The development of methods for multi-dimensional classification of (nano)particles requires a uniform mathematical framework for the quantitative evaluation of fractionation processes of particles of any size, shape and property. Moreover, this general framework needs to be applicable to any classification process. In our project within PP 2045, we developed such a universal mathematical framework by extending the traditional approach for describing one-dimensional PSDs and separation processes to the general multi-dimensional case. Key parameters of our multi-dimensional approach are a particle property vector $\vec{x}$ and multi-dimensional separation efficiency functions depending on $\vec{x}$. The particle property vector is a function of the particle dimensions and, therefore, accounts for the more complex properties of non-spherical particles. $\vec{x}$ is not restricted to the particle dimensions, as it can be applied to quantify the distribution of any property of a particle ensemble (for example, composition and size distributions of alloys and minerals and property distributions like band gap, electric or thermal conductivity).

To make the mathematical framework more illustrative, we applied it to the fractionation of gold nanorods with lognormal distribution in length and diameter using the following separation processes:

1. A one-dimensional separation regarding the volume (for example, via centrifugation), which splits the feed PSD into two fractions (coarse and fine), each with a two-dimensional PSD;
2. Separation regarding sedimentation properties and electrophoretic mobility via two orthogonal processes, which results in four two-dimensional size fractions.

For both cases, we demonstrated that the PSDs of the size fractions and the relative amounts of particles in each fraction are accessible from the feed PSD and the separation efficiency functions. Thus, if the feed PSD and separation efficiency functions are known, the outcome of multi-dimensional fractionation processes can be calculated. For example, in our previous work, we showed that a Rietema hydrocyclone should enable shape-classification of spherical (2–7 µm in diameter) and plate-like silica particles of the same volume [38,78]. As plate-like particles exhibit a larger projection area than spheres, the plates experience higher drag forces. Therefore, it is expected that the plate-like silica

particles are enriched in the coarse fraction of the hydrocyclone, whereas the spherical silica particles are expected in the fine fraction [38]. Moreover, we modeled the classification regarding particle size and density in a disk separator and expected to find the small and low-density particles in the fine fraction [38].

Another important precondition for establishing multi-dimensional particle separation processes is fast and accurate methods for measuring multi-dimensional particle size and property distributions. Traditional image analysis of micrographs from optical or electron microscopy provides data regarding size and shape but is very tedious as some hundreds or even thousands of particles have to be evaluated to provide the sufficient statistical significance of the result. Thus, fast ensemble methods for measuring the particle size and property distributions in suspension are required as they provide better statistics than imaging methods. We demonstrated that MWL-AUC is the gold standard for multi-dimensional nanoparticle characterization in suspension as it provides the sedimentation and diffusional properties of a settling particle in combination with its UV/Vis/NIR-spectrum [42,47,48,64]. For the case that the UV/Vis/NIR-spectra can be simulated using a physical model under consideration of the sedimentation properties of the settling species, two-dimensional particle size/property distributions are accessible from MWL-AUC data using the optical back coupling method [38–40,42]. As physical models for describing the surface plasmon resonance of gold nanoparticles are available, we were able to measure the length and diameter distribution of different commercial gold nanorod samples via the MWL-AUC optical back coupling method [38–40,42]. We found the measured two-dimensional size distributions to be in excellent agreement with the PSDs derived from statistical TEM image analysis, giving evidence for the high accuracy of the optical back coupling method. Moreover, we compared the performance of different physical models for the surface plasmon resonance of gold nanorods (classical Gans theory, FEM simulations and a longitudinal polarization model) with each other and found that the three models give almost identical distributions for the lengths and diameters of the gold nanorods [38–40,42]. The optical back coupling method is not limited to gold nanorods. We applied it for measuring the size distribution of gold bipyramids from the group of Liz-Marzan [41] and for the quantitative analysis of mixtures of differently shaped gold nanoparticles (gold nanorods and spherical nanoparticles) [42]. For the latter, we were able to quantify the relative amounts of gold nanorods and spherical gold nanoparticles and measure simultaneously the PSDs of the spheres and the length and diameter distributions of the gold nanorods [42]. In addition, the optical back coupling method is well established for measuring the PSD and composition distribution of spherical gold/silver alloy nanoparticles [59,79].

For $MoS_2$ nanosheets, mean values for the lateral diameter $L$ and the number of layers $N$ are accessible from the UV/Vis-spectra [63]. We estimated distributions for $L$ and $N$ for a $MoS_2$-nanosheet sample via the size classification of the sample via the preparative centrifugation and spectroscopic evaluation of each size fraction. The obtained distributions reveal that few-layer structures with lateral diameters around 100 nm were formed. The $MoS_2$-nanosheet dimensions are comparable with the dimensions of few-layer-graphene particles from stirred media delamination [44,45]. It should be noted that the preparative centrifugation method gave estimates for the $L$ and $N$ distributions of the feed sample. In future work, we will determine the two-dimensional PSDs of $MoS_2$-nanosheets more precisely via MWL-AUC.

## 5. Conclusions

We developed a universal framework for describing multi-dimensional PSDs and the quantitative evaluation of multi-dimensional particle classification processes by extending the traditional one-dimensional approaches. We show how multi-dimensional distributions can be converted into each other. Examples are given for the important case of lognormal PSDs. If the conversion is multiplicative (in the case of nanorods from number-based to volume-based PSDs), the PSDs remain lognormal with the covariance conserved. If the conversion is additive (in the case of nanorods from number-based to surface-based PSDs),

the conversion leads to the sum of lognormal PSDs. This framework was used to predict the outcomes of one-dimensional and two orthogonal separation processes applied to gold nanorods.

MWL-AUC in general, and in particular the optical back coupling method, provides a fast and highly accurate ensemble method for measuring two-dimensional PSDs in suspension. The optical back coupling method has versatile application under the precondition that physical models for the simulation of the UV/Vis-spectra of the settling species exist. We demonstrated this method for plasmonic gold nanorods. The framework for describing multi-dimensional PSDs and separation processes is the basis for developing multi-dimensional separation processes for nanoparticles. Fast and reliable measurement techniques such as MWL-AUC are a key requirement for the understanding, design and scale-up of procedures to produce and formulate multivariate particle systems.

**Supplementary Materials:** The following supporting information can be downloaded via this link: https://www.mdpi.com/article/10.3390/powders3020016/s1: pdf-file "Peukert-SI1-PP2045": S1: Code for creating the figures; S2: Details regarding multivariate normal and lognormal distributions; S3: Weighting particle size distributions differently; excel-file "Peukert-SI2-PP2045": Raw data of the results shown in Figures 5–7 (UV/Vis spectra of the $MoS_2$ fractions and AUC data of the Au nanorod sample).

**Author Contributions:** Conceptualization, C.D. and W.P.; methodology, D.L. and J.W.; software, D.L. and J.W.; validation, J.W.; formal analysis, J.W. and W.P.; investigation, C.D. and D.L.; resources, W.P.; data curation, C.D. and D.L.; writing—original draft preparation, C.D.; writing—review and editing, C.D, D.L., J.W. and W.P.; visualization, D.L. and C.D.; supervision, W.P.; project administration, J.W., D.L. and C.D.; funding acquisition, W.P. All authors have read and agreed to the published version of the manuscript.

**Funding:** This study was funded by the Deutsche Forschungsgemeinschaft (DFG, German Research Foundation) within priority program PP 2045 "Highly specific and multidimensional fractionation of fine particle systems with technical relevance" (Project-ID 313858373).

**Institutional Review Board Statement:** Not applicable, as our studies did not involve humans or animals.

**Informed Consent Statement:** Not applicable, as our studies did not involve humans.

**Data Availability Statement:** The data that support the findings of this study are available in the Supplementary Materials.

**Acknowledgments:** We gratefully acknowledge financial support from Deutsche Forschungsgemeinschaft (PP 2045) and Friedrich-Alexander-Universität Erlangen-Nürnberg within the funding program "Open Access Publication Funding". The authors greatly thank Nabi Traore for the TEM investigation of the gold nanorods and Hadi Soltanmoradi for performing the AUC measurement on the gold nanorods. We further acknowledge the support of the collaborative research centers CRC 953 (Project-ID 182849149) and CRC 1411 (Project-ID 416229255) at FAU.

**Conflicts of Interest:** The authors declare no conflicts of interest. The funders had no role in the design of the study; the collection, analyses, or interpretation of data; the writing of the manuscript; or the decision to publish the results.

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
