# Peer review of "Size and Shape Selective Classification of Nanoparticles"

_2674-0516, doi:10.3390/powders3020016_

Round 1
Reviewer 1 Report
Comments and Suggestions for Authors
The manuscript presents a methodology, based on experiments and simulations, to classify and separate particles based on their size and shape. Typically, particles are described in terms of their size distribution, which can be measured by various methods, including inversion of scattering data, image analysis, single particle (Coulter) counting methods, and analytical centrifugation. A description of more than one feature of particles, for example, shape and size, is seldom done in the literature due to the difficulty in performing such a description and the even more significant challenge in achieving this characterization. This work goes beyond the state of the art and presents a comprehensive approach based on multiple lognormal distributions, aiming at characterizing particles using more than one feature. The method is tested with gold nanorods, described in terms of length and aspect ratio, and molybdenum disulfide nanosheets. The authors first develop a general theoretical treatment of multiple cumulative distribution functions, then use it to create intermediate distributions, such as aspect ratio distribution and specific surface area distribution. Then, actual nanoparticle suspensions were separated using analytical ultracentrifugation, and the fractions were characterized using UV-VIS spectroscopy.
The work is well-written, clear, and comprehensive and provides a powerful new nanoparticle-characterization strategy. The results are solid and discussed comprehensively. Publication is recommended without any remarks.
Author Response
The authors would like to thank the reviewer for this very positive feedback. It shows that the methods presented in our paper are important for further development of multi-dimensional separation methods.
Reviewer 2 Report
Comments and Suggestions for Authors
In this study proposed a uniform framework for the quantitative description of multidimensional particle property distributions and of multi-dimensional fractionations as well as fast and highly accurate ensemble methods for measuring multi-dimensional particle property distributions. In summary, upon careful modifications, the article may be considered for acceptance. Detailed comments are as follows:
1. What instruments are used to separate particles of different sizes?
2. How efficient is the separation?
3. Is it possible to separate nanomaterials below 10 nm, such as quantum dots?
4. Is it possible to separate different AuNRs, such as pentagon, hexagonal, heptagon?
Author Response
We extended the introduction section in the revised manuscript to provide answers to your questions 1-3. Generally, all separation methods mentioned in the introduction (chromatography, size-selective precipitation, (gel) electrophoresis, field-flow fractionation, centrifugation and magnetic-particle assisted methods) are useful for classification of particles < 100 nm regarding size. In particular, nanoparticle chromatography, analytical ultracentrifuges providing sufficiently high g-values and size-selective precipitation (for semiconductor quantum dots) enable separation of particles < 10 nm. The efficiency of the separation methods is included in the literature discussion. We have shown in earlier publications, that the separation efficiencies defined as (x75-x25)/x50 are well above 0.5 and often reach values of 0.9, with 1 being a perfect separation.
Recently we have shown that a two-dimensional analysis via analytical ultracentrifugation can also be applied to 5-sided double pyramids (cited as reference 41 in the manuscript). Question 4 is in general, however, quite challenging for separation of non-spherical particles with identical volume and aspect ratio, but different geometry. Separation is only possible if the determining physical parameters of the particle ensemble differ sufficiently. Centrifugation will hardly work because this method classifies regarding the volume and the frictional ratio of the particles which is a function of the aspect ratio. Thus, centrifugation is only efficient for separation of particles which differ either in volume or in the aspect ratio. Key parameter is a sufficiently varied sedimentation coefficient as function of the particle geometry. Size exclusion chromatography will only work for particles with different diffusion coefficients which depend on the hydrodynamic diameter. Interaction chromatography is expected to work for differently shaped crystals as different geometries come along with different crystal faces. The surface energy and therefore the interaction behaviour of different crystal faces can be different and thus used to tune the interactions with the stationary phase.
In conclusion, this question is exciting for future research, a general answer is beyond the focus of this paper.
Reviewer 3 Report
Comments and Suggestions for Authors
In this paper, the authors introduced a framework to describe the multi-diemsional PSDs. The research is comprehensive and they proved that the method is practical by applying it to the experimental data collected. But just one comment that should be addressed before publication.
The novelty of this paper is not very clear and strong sounded. Authors mentioned several multi-dimensional particle fractionation techniques have been developed already. What’s the advantage of your technique over theirs? Why do you think yours are important? Can you include this information in the introduction part?
Author Response
Several novel methods for multi-dimensional particle fractionation were developed within the German priority program PP 2045. In the beginning of the research activities in PP 2045 it was not clear, how to describe multi-dimensional property distributions and fractionations quantitatively. In our project in PP 2045 we developed a uniform mathematical framework for describing multi-dimensional normal-distributed PSDs and fractionations by extending the well-known approaches for one-dimensional PSDs and separations to the multi-dimensional case. In this paper we extended our framework to multi-dimensional lognormal PSDs and multi-dimensional fractionations of particles with lognormal distribution of different particle dimensions. This extension is important because PSDs are often lognormally distributed. For illustration we model the fractionation of gold nanorods with lognormal distribution in length and diameter.
This paper is the final report of our project in PP 2045. We revised the introduction of our paper to provide a clearer discrimination of new results presented in this paper from already published results of our project.